# Fault Diagnosis for Rolling Bearings Based on Multiscale Feature Fusion Deep Residual Networks

**Xiangyang Wu** [1,2], **Haibin Shi** [3,*] and **Haiping Zhu** [3,*]

1 School of Mechanical Engineering, Southwest Jiaotong University, Chengdu 611756, China
2 CRRC Qingdao Sifang Rolling Stock Co., Ltd., Qingdao 266111, China
3 School of Mechanical Science and Engineering, Huazhong University of Science and Technology, Wuhan 430074, China
* Correspondence: haibinshi11@163.com (H.S.); haipzhu@hust.edu.cn (H.Z.)

**Abstract:** Deep learning, due to its excellent feature-adaptive capture ability, has been widely utilized in the fault diagnosis field. However, there are two common problems in deep-learning-based fault diagnosis methods: (1) many researchers attempt to deepen the layers of deep learning models for higher diagnostic accuracy, but degradation problems of deep learning models often occur; and (2) the use of multiscale features can easily be ignored, which makes the extracted data features lack diversity. To deal with these problems, a novel multiscale feature fusion deep residual network is proposed in this paper for the fault diagnosis of rolling bearings, one which contains multiple multiscale feature fusion blocks and a multiscale pooling layer. The multiple multiscale feature fusion block is designed to automatically extract the multiscale features from raw signals, and further compress them for higher dimensional feature mapping. The multiscale pooling layer is constructed to fuse the extracted multiscale feature mapping. Two famous rolling bearing datasets are adopted to evaluate the diagnostic performance of the proposed model. The comparison results show that the diagnostic performance of the proposed model is superior to not only several popular models, but also other advanced methods in the literature.

**Keywords:** deep learning; residual learning; multiscale feature fusion deep residual networks; feature fusion; intelligent fault diagnosis

## 1. Introduction

The development of digitalization and intellectualization puts forward high requirements for the reliability of mechanical equipment [1–3]. Because of long device running times, it is inevitable that cracks, corrosion or other faults will occur in rolling bearing operations under high temperatures, high pressures and other harsh environments. Therefore, timely and accurate fault diagnosis of rolling bearings is necessary for mechanical equipment, diagnoses which can effectively avoid further deterioration of mechanical faults, and even serious accidents and huge economic losses [4–6].

Currently, the waveform signal is the most widely used monitoring signal for the fault diagnosis of rolling bearings [7–9]. Multidimensional features in time-domain, frequency-domain and time-frequency domain are widely extracted for signal processing. Cheng et al. [10] adopted 12 different time-domain statistical features to indicate the health status of rolling bearings. Betta et al. [11] adopted the fast Fourier transform to extract frequency-domain features from raw signals. Zheng et al. [12] introduced a spectral envelope-based monitoring signal processing method for fault diagnosis. Bouzida et al. [13] implemented the discrete wavelet transform for extracting information from signals of a wide range of frequencies, achieving the fault diagnosis of induction machines. Yu et al. [14] used empirical mode decomposition to convert the raw signals to the local Hilbert marginal spectrum, which is utilized to extract the time-frequency

domain features for the fault diagnosis of roller bearings. However, such methods rely heavily on expert knowledge and experience, which restricts their application in complex practical scenarios.

With the rapid progress of intelligent sensing and computer technology, artificial intelligence-based fault diagnosis approaches have become a research hotspot [15–19]. Among all artificial intelligence methods, the machine learning method is most prominent, since it can adaptively capture potential data features in monitoring signals without too much expert knowledge and experience. Li et al. [20] proposed a Bayesian network-based fault diagnosis method, and applied it as to the motor bearing. Yang et al. [21] constructed a support vector machine model integrated with an intrinsic mode function envelope spectrum for fault diagnosis with few training samples. Boutros et al. [22] detected and diagnosed the faults of a bearing and cutting tool based on hidden Markov models, and achieved more than 95% classification accuracy on both objects. Although AI-based fault diagnosis methods have achieved outstanding results to some extent, they gradually lose their dominant position in complex diagnosis tasks with the booming of industrial big data, since their shallow architectures cannot effectively capture the many potential data features within massive data.

Because of the ability to adaptively capture and extract high-dimensional information from massive monitoring data, deep learning (DL) is widely utilized in the field of fault diagnosis [23–26]. Some DL methods, such as deep neural networks (DNN) [27], the deep Boltzmann machine [28], recurrent neural networks [29] and deep autoencoders [30] and convolutional neural networks (CNN) [31,32], have shown their prominent capabilities and been successfully applied. Among these DL models, CNN shows the most outstanding feature capture capability due to its unique convolution and pooling structure. Li et al. [33] built a CNN model for the fault diagnosis of rolling bearings, and validated this method on these different datasets. Xia et al. [34] utilized CNN to fuse multi-sensor signals, and successfully achieved the fault diagnosis of bearings. Lu et al. [35] proposed privacy-preserving federated learning framework by using CNN as the backbone network, and applied it on the fault diagnosis of rolling bearings.

Although DL has become a popular method, there are two common problems in the DL-based fault diagnosis methods:

(1) Currently, many researchers attempt to deepen the layers of the DL model for better nonlinear feature extraction ability and higher diagnostic accuracy. With the deepening of the network layers and the expanding of the parameter scale, degradation problems often occur in the DL model training process. Specifically, traditional DL models require the utilization of back-propagation algorithms for pass errors, layer by layer. With the increase of nonlinear layers, the influence of gradient disappearance or explosion will gradually increase, which means the gradient tends to the extreme value (maximum or minimum), making the optimization process more and more difficult. This makes it difficult for the training errors to continue to decline when the training is reduced to a certain extent. Guo et al. [36] constructed a deep CNN model for the fault diagnosis of rolling bearings. However, the convergence was quite slow and the training process required thousands of epochs. Zhu et al. [37] proposed a deep autoencoder-based fault diagnosis method and achieved excellent performance on rolling bearings. However, model training took thousands of cycles to complete in both of the two experimental cases, which greatly increases the challenge of this model in practical engineering applications. Fortunately, residual learning, as a new extension of DL with special skip connection structure, offers a promising solution to the degradation problem.

(2) The second problem is that most researchers neglect the use of multiscale features; this makes the extracted data features lack diversity. Jing et al. [38] proposed a fault-diagnosis method of rolling bearings based on the CNN model, where multiple convolutional layers and pooling layers stack to form a depth model. Lu et al. [39]

constructed a CNN model for the fault diagnosis of rolling bearings, but it did not consider the extraction of multiscale features.

To overcome these problems, this paper introduces a novel multiscale feature fusion deep residual network (MFFDRN) for the fault diagnosis of rolling bearings, which contains multiple multiscale feature fusion blocks (MFF blocks) and a multiscale pooling layer (MPL). The MFF block is designed to automatically extract the multiscale features from raw signals, and further compress them for higher dimensional feature mapping. The stacking of multiple MFF blocks enables MFFDRN to capture and extract more abstract and high-dimensional features from raw signals. Then, MPL is constructed to fuse the extracted multiscale feature mapping.

The main contributions are summarized as follows:

(1) An end-to-end fault diagnosis approach based on residual learning is proposed with enhanced feature extraction ability, one which can effectively extract potential features from 1-D raw signals without handcrafted feature extraction.
(2) A novel MFF block is designed to automatically extract, fuse and compress the multiscale features. This structure can extract multiscale features with fewer filter channels.
(3) A new multiscale pooling method is proposed to broaden the receptive field of MFFDRN.

The rest of this paper is arranged as follows. The proposed MFFDRN approach is introduced in detail in Section 2. Section 3 introduces two experimental cases. Finally, Section 4 sets forth a conclusion.

## 2. Proposed MFFDRN

### 2.1. CNN

CNN is developed on the basis of feedforward neural networks, which use the mechanism of local connection and weight sharing to reduce the number of network parameters. Therefore, the training time of a CNN model is much shorter than that of an ANN model with the same number of parameters. A CNN model usually contains a convolutional layer, an activation layer and a pooling layer.

The convolutional layer outputs deeper feature maps through the convolution between pooling kernels and input feature maps; different sizes of convolutional kernels will lead to different convolution results. The convolution operation is expressed as

$$y^k = w^k \otimes x + b_c \tag{1}$$

where $y^k$ represents the convolution result of the $k$th channel, $w^k$ denotes the $k$th convolutional kernel, $\otimes$ is the convolution operator, $x$ indicates the input feature map and $b_c$ represents the bias item.

To increase the nonlinearity of CNN, an activation function is applied to activate the feature maps output by convolutional layers. Rectified linear unit (ReLU) is commonly used in CNNs, for it usually learns much faster than other activation functions [40]. The definition of ReLU is shown as

$$g(z) = \max\{0, z\} \tag{2}$$

"Pooling layers" is adopted to diminish the dimension of input matrix, which uses the overall statistical value of adjacent data at a certain location as the output at the same position. Compared with maximum pooling, average pooling can retain more local information of the input data, therefore is often integrated in CNN models, expressed as

$$p = \psi down(y) + b_p \tag{3}$$

where $\psi$ represents the multiplicative bias term, $down(\cdot)$ is the operation of average pooling, $y$ indicates the input matrix, and $b_p$ is the additive bias.

### 2.2. Residual Learning Module

Due to the nonlinear mapping of CNN layers, the output features of each layer will lose some information relative to the input features. With the deepening of the network, the impact of this phenomenon will become more and more serious, leading to the degradation problem of deep CNN in the training process, that is, model training becomes very difficult and the training accuracy of the network reaches saturation or even decreases gradually. To address the degradation problem, residual learning was designed with the special skip connection structure. In this paper, residual learning block (ResBlock) is adopted in the proposed MFFDRN.

The structure of the ResBlock constructed in MFFDRN is shown in Figure 1. A ResBlock includes two convolutional blocks (ConvBlocks), each of which contains a convolutional layer, a batch normalization layer (BatchNorm) and a ReLU activation layer. The concatenation of two ConvBlocks can effectively improve the capabilities of data capture and feature mining. In addition, the skip connection structure allows the output data to contain information about all input data to alleviate the degradation problem in the deep learning training. The mathematical expression of ResBlock is shown as

$$H(X) = f(X) + X \tag{4}$$

where $H$ denotes the nonlinear transformation process of ConvBlocks, $X$ represents the input data, and $f$ represents the transformation in stacked ConvBlocks.

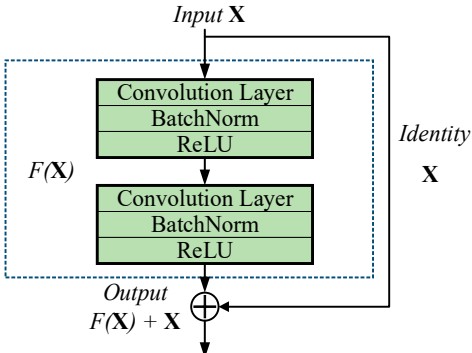

**Figure 1.** Residual learning block.

The batch normalization layer can solve the internal covariate shift problem during training iteration by normalizing input data.

The batch normalization layer (BatchNorm) is for the normalization of the input data in ResBlocks, which can solve the problem of internal covariate shift in the training [41]. The process of BatchNorm can be expressed as follows,

$$\begin{cases} \mu = \frac{1}{N} \sum_{i=1}^{N} x_i \\ \sigma^2 = \frac{1}{N} \sum_{i=1}^{N} (x_i - \mu)^2 \\ \hat{x}_i = \frac{x_i - \mu}{\sqrt{\sigma^2 + \epsilon}} \\ y_i = \mu \hat{x}_i + \beta \end{cases} \tag{5}$$

where $x_i$ and $y_i$ are the independent variable and the dependent variable of the $n$th observation in a mini-batch of size $N$, $\mu$ and $\beta$ are variables learned to scale and shift distributions. $\epsilon$ is a positive constant close to zero to make the denominator always positive.

### 2.3. Proposed MFFDRN Architecture

Figure 2 shows the architecture of the MFFDRN. We can see that MFFDRN contains an initial ConvBlock, 3 MFF Blocks, a MPL with 3 multiscale pooling blocks (MSPs) and

a fully-connected layer with softmax as the activation function. In addition, the MFFDRN model uses raw vibration signal as input without manual feature extraction and selection.

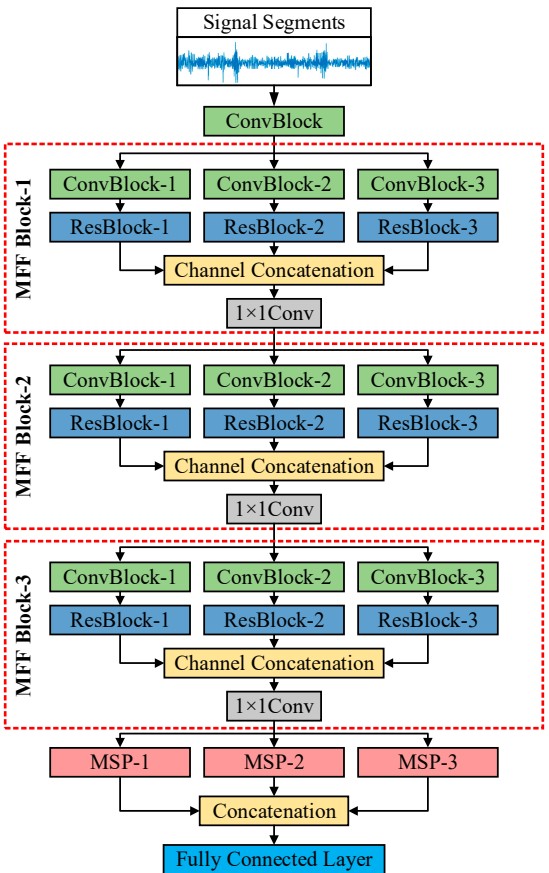

**Figure 2.** Structure of MFFDRN.

MFF Block is the core part of MFFDRN, which includes 3 ConvBlocks, 3 ResBlocks and a bottleneck layer with kernel size of $1 \times 1$ ($1 \times 1$ convolution). The difference between the 3 ConvBlocks is the kernel size, similar to the ResBlocks. To extract multiscale features from raw signal, feature maps output from various Resblocks are re-concatenated by channel concatenation. Then, the multiscale features pass through a convolutional layer with kernel size of $1 \times 1$ for the fusion of feature maps, so as to diminish feature map channels without losing information. These components enable the MFF Block to have the ability of multiscale features fusion and enhance the IDF performance of MFFDRN. Due to the utilization of MFF blocks, the feature map channels in MFFDRN are much fewer in number than those in CNN. Next, a MPL is utilized to mine the most effective feature information from the output feature maps of MFF blocks. It can be seen in Figure 3 that each MSP includes a convolutional layer with $1 \times 1$ kernels and an average pooling layer. For features with different scales, the hyperparameters of MSPs are various.

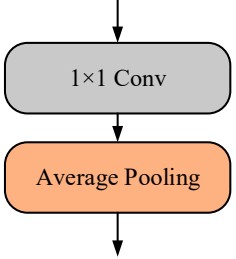

**Figure 3.** Structure of multiscale pooling block.

The configuration of MFFDRN is listed in Table 1. L indicates the length of input signal segments and T denotes the number of fault types; $4 \times L \times 1$ represents that the feature maps have four channels and their sizes are $L \times 1$; $C3 \times 1$ denotes that the convolutional kernel size is $3 \times 1$; S1 denotes that stride is 1 and $2 \times (C3 \times 1, S1)$ means that the parameters of the two ConvBlocks that make up of the ResBlock are both $C3 \times 1$; $P[L/16 \times 1]$ denotes that the pooling kernel size is one-sixteenth of the length of feature maps; CAT represents channel concatenation. It is noted that strides are set to 1 by default in all the convolutional layers.

**Table 1.** The configuration of MFFDRN.

| Block Name | Component | Parameter | Output Size |
|---|---|---|---|
| Input | - | - | $1 \times L \times 1$ |
| ConvBlock | - | - | $4 \times L \times 1$ |
| MFF Block-1 | ConvBlock-1 | $C3 \times 1, S1$ | $8 \times L \times 1$ |
| | ConvBlock-2 | $C7 \times 1, S1$ | $8 \times L \times 1$ |
| | ConvBlock-3 | $C11 \times 1, S1$ | $8 \times L \times 1$ |
| | ResBlock-1 | $2 \times (C3 \times 1, S1)$ | $8 \times L \times 1$ |
| | ResBlock-2 | $2 \times (C7 \times 1, S1)$ | $8 \times L \times 1$ |
| | ResBlock-3 | $2 \times (C11 \times 1, S1)$ | $8 \times L \times 1$ |
| | CAT | - | $24 \times L \times 1$ |
| | $1 \times 1$ Conv | $C1 \times 1, S1$ | $8 \times L \times 1$ |
| MFF Block-2 | ConvBlock-1 | $C3 \times 1, S1$ | $16 \times L \times 1$ |
| | ConvBlock-2 | $C7 \times 1, S1$ | $16 \times L \times 1$ |
| | ConvBlock-3 | $C11 \times 1, S1$ | $16 \times L \times 1$ |
| | ResBlock-1 | $2 \times (C3 \times 1, S1)$ | $16 \times L \times 1$ |
| | ResBlock-2 | $2 \times (C7 \times 1, S1)$ | $16 \times L \times 1$ |
| | ResBlock-3 | $2 \times (C11 \times 1, S1)$ | $16 \times L \times 1$ |
| | CAT | - | $48 \times L \times 1$ |
| | $1 \times 1$ Conv | $C1 \times 1, S1$ | $16 \times L \times 1$ |
| MFF Block-3 | ConvBlock-1 | $C3 \times 1, S1$ | $32 \times L \times 1$ |
| | ConvBlock-2 | $C7 \times 1, S1$ | $32 \times L \times 1$ |
| | ConvBlock-3 | $C11 \times 1, S1$ | $32 \times L \times 1$ |
| | ResBlock-1 | $2 \times (C3 \times 1, S1)$ | $32 \times L \times 1$ |
| | ResBlock-2 | $2 \times (C7 \times 1, S1)$ | $32 \times L \times 1$ |
| | ResBlock-3 | $2 \times (C11 \times 1, S1)$ | $32 \times L \times 1$ |
| | CAT | - | $96 \times L \times 1$ |
| | $1 \times 1$ Conv | $C1 \times 1, S1$ | $32 \times L \times 1$ |
| MSP-1 | $1 \times 1$ Conv | $C1 \times 1, S1$ | $1 \times L \times 1$ |
| | Average Pooling | $P[L/16 \times 1], S[L/16 \times 1]$ | $1 \times L/16 \times 1$ |
| MSP-2 | $1 \times 1$ Conv | $C1 \times 1, S1$ | $4 \times L \times 1$ |
| | Average Pooling | $P[L/8 \times 1], S[L/8 \times 1]$ | $4 \times L/8 \times 1$ |
| MSP-3 | $1 \times 1$ Conv | $C1 \times 1, S1$ | $8 \times L \times 1$ |
| | Average Pooling | $P[L/4 \times 1], S[L/4 \times 1]$ | $8 \times L/4 \times 1$ |
| Fully Connected Layer | - | - | $T \times 1$ |

## 3. Experimental Study

To evaluate the effectiveness and the generalization ability of the proposed MFFDRN in rolling bearing fault diagnosis, two cases were studied using two different famous datasets, i.e., the Paderborn University bearing dataset [42] and the dataset from the Society for Machinery Failure Prevention Technology (MFPT dataset) [43].

Two evaluation indicators were used to evaluate the fault diagnosis performance, including accuracy and the macro F1-score ($F$), respectively, expressed as

$$accuracy = \frac{a}{A} \tag{6}$$

$$F = \frac{1}{n} \cdot \sum_{i=1}^{n} \left( 2 \cdot \frac{p_i \cdot r_i}{p_i + r_i} \right) \tag{7}$$

where $a$ is the number of test samples correctly diagnosed. $A$ represents the total number of test samples. $n$ indicates the number of all fault types. $p_i$ and $r_i$ denote precision and recall of $i$-th fault type, respectively.

The training and testing of all models were implemented by Pytorch 1.7 on a workstation with a Windows 10 operation system and TITAN XP GPU.

*3.1. Case One*

3.1.1. Data Description

This dataset was obtained from a modular test rig shown in Figure 4. Tested rolling bearings had three kinds of conditions: healthy bearings, bearings with an inner race fault and bearings with an outer race fault. Additionally, two types of bearings damage were used in these experiments: the artificial damages and the real damages from accelerated lifetime tests. As presented in Table 2, the experiments were carried out under four different operating conditions with various rotating speeds, load torques, and radial forces applied to the bearings. In this paper, only the bearings with real damages from accelerated lifetime tests were adopted to better evaluate the diagnostic performance of MFFDRN in real industrial applications.

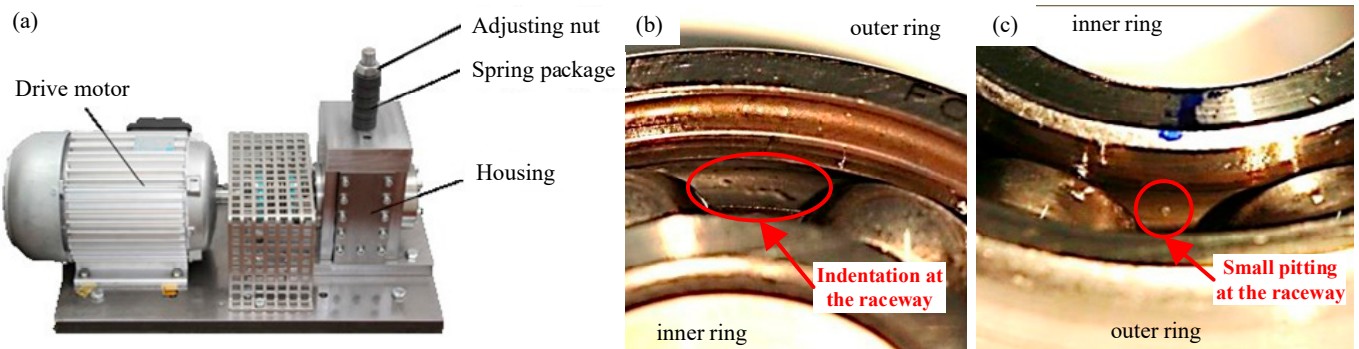

**Figure 4.** (**a**) Test rig in Case 1; (**b**) outer ring fault; and (**c**) inner ring fault.

**Table 2.** Operating parameters of test rig in Case 1.

| NO. | Rotating Speed [rpm] | Load Torque [Nm] | Radial Force [N] | Name of Settings |
|-----|----------------------|------------------|------------------|------------------|
| 0 | 1500 | 0.7 | 1000 | N15_M07_F10 |
| 1 | 900 | 0.7 | 1000 | N09_M07_F10 |
| 2 | 1500 | 0.1 | 1000 | N15_M01_F10 |
| 3 | 1500 | 0.7 | 400 | N15_M07_F04 |

The procedure of vibration signal preprocessing is shown in Figure 5. The signals are divided into small pieces with the length of 5120 as input sample. No overlay exists in the process of signal segmentation. The final signal segments under various health conditions are exhibited in Figure 6.

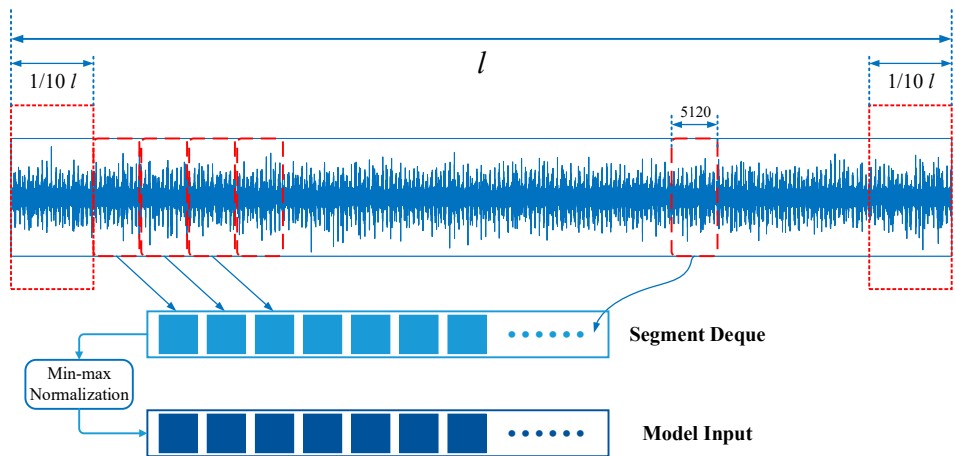

**Figure 5.** Schematic diagram of data preprocessing.

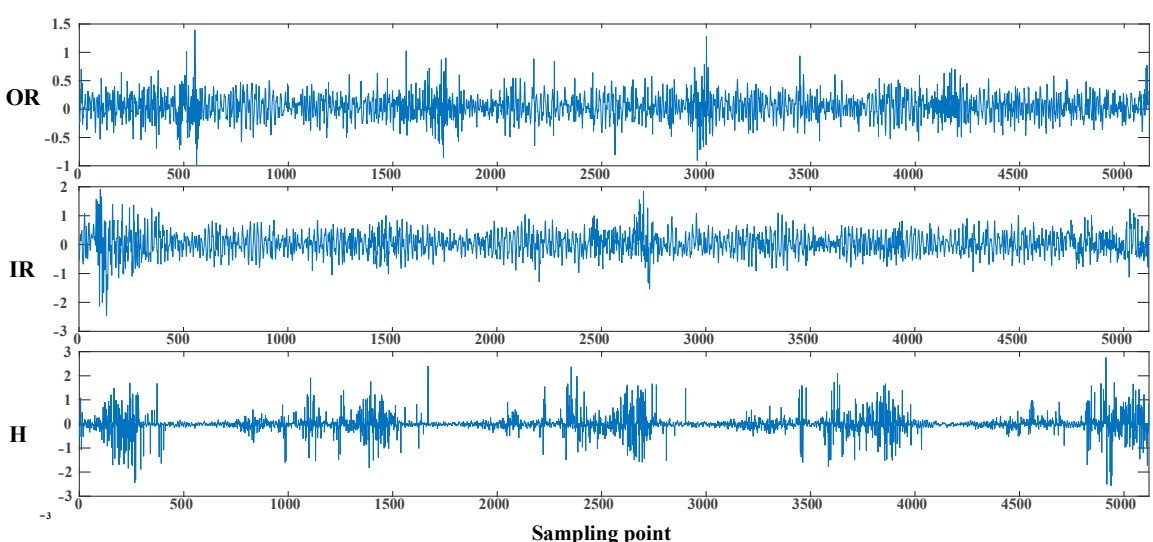

**Figure 6.** Signal segments with different health conditions in Case 1. OR denotes outer race fault, IR denotes inner race fault and H represents the healthy sample.

Next, the min-max normalization is adopted to normalize the raw signal segment, defined as

$$\widetilde{x} = \frac{x - \min(x)}{\max(x) - \min(x)} \tag{8}$$

where $x$ represents the raw data, and $\max(x)$ and $\min(x)$ are the maximum and minimum in the signal segment.

A total of 48,052 signal samples were finally obtained for IDF, 80% of which were randomly selected for training models, and the rest were utilized as test data samples.

### 3.1.2. Results Comparison and Analysis

To verify the superiority of the proposed approach, the comparison between MFFDRN and several popular methods were implemented, such as DNN, CNN and single scale deep residual networks (DRN) with kernel sizes of $3 \times 1$, $7 \times 1$ and $11 \times 1$ (shown as DRN-3, DRN-7 and DRN-11). The structure of DRN degenerates from MFFDRN; the differences are that the DRN is a single-scale network, the $1 \times 1$ convolutional layer is removed and the global average pooling layer is utilized as the multiscale pooling layer. The experiment was repeated five times. The training settings for all models are shown in Table 3. All models

used the same training and testing set. Additionally, the dataset was re-divided randomly after each experiment. The diagnostic results are illustrated in Figure 7 and Table 4.

**Table 3.** The training settings for all models.

| Model Settings | Value |
|---|---|
| Epoch number | 40 |
| Optimizer | Adam |
| Initial learning rate | 0.001 |
| Batch size | 16 |
| Regularization | L2 regularization in convolutional layers (weight as 0.00001) |

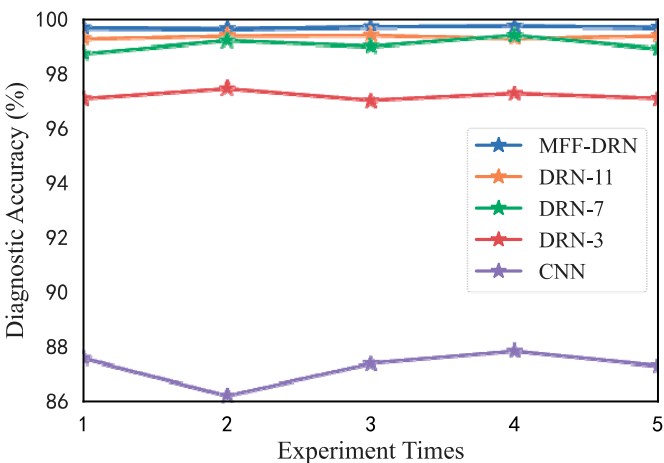

**Figure 7.** Diagnostic accuracy comparison of five methods in Case 1.

**Table 4.** Experimental results in Case 1 (%).

| Model | Max Acc | Min Acc | Mean Acc | SD | Mean *F* | Average Training Time (s) | Average Testing Time per Sample (s) |
|---|---|---|---|---|---|---|---|
| DNN | 66.31 | 63.99 | 65.26 | 0.868 | 64.65 | 14,683.94 | 0.31 |
| CNN | 87.88 | 86.22 | 87.29 | 0.566 | 87.27 | 23,908.26 | 0.52 |
| DRN-3 | 97.48 | 97.03 | 97.21 | 0.166 | 97.22 | 25,438.37 | 0.55 |
| DRN-7 | 99.43 | 98.75 | 99.07 | 0.234 | 99.08 | 26,876.36 | 0.61 |
| DRN-11 | 99.45 | 99.31 | 99.39 | 0.055 | 99.36 | 28,457.64 | 0.63 |
| MFFDRN | 99.78 | 99.68 | 99.73 | 0.035 | 99.72 | 82,359.04 | 0.75 |

It is obvious from Figure 7 that MFFDRN achieved the highest diagnosis accuracy in each experiment and was quite stable in its prediction results; the accuracies of DRN-11, DRN-7 and DRN-3 were lower and less stable than that of MFFDRN. The prediction of CNN was worse than the models mentioned above. The DNN is not presented in the picture because its performance is much worse than other models.

Table 4 shows the maximum, minimum and mean accuracy, standard deviation (SD) of accuracy and mean F1-score (mean *F*) of DNN, CNN, DRN-3, DRN-7, DRN-11 and MFFDRN. The MFFDRN has the highest performance of all the indicators. The detailed discussions about the result comparison are summarized as follows.

(1) Among these models, the DNN has the worst performance, which is due to the relatively shallow network structure of DNN.
(2) CNN, DRNs and the MFFDRN are much better than DNN. This demonstrates the good data mapping ability of the convolution operation.
(3) The performance of three DRNs is positively correlated with filter sizes and much greater than CNN. It indicates the advantage of residual learning and shows that bigger filters have better feature-mapping abilities.

(4)  MFFDRN has the highest indicators on max accuracy, min accuracy and mean accuracy, and the smallest standard deviation. In addition, the mean *F* of MFFDRN is the highest. This is because the multiscale extraction structure enabled MFFDRN has the enhanced feature extraction ability. In addition, the feature fusion structure of MFFDRN can effectively fuse multiscale features to obtain better diagnostic performance.

(5)  In terms of time consumption, MFFDRN shows no obvious advantages compared with other models. It is because the MFFDRN has a relatively complex network structure. The average testing time of all the models meets the industrial requirements, which proves the MFFDRN can be applied to practical equipment in industrial systems.

In order to better understand the diagnosis results, the results of the last experiment will be shown in detail. The classification accuracies of each fold in the four-fold cross-validation test for MFFDRN were 99.54%, 99.75%, 99.79% and 99.69%, with an average value of 99.69%. The classification results of testing samples for six models are shown in Figure 8. It can be seen that MFFDRN can better diagnose the faults of rolling bearings than can other models.

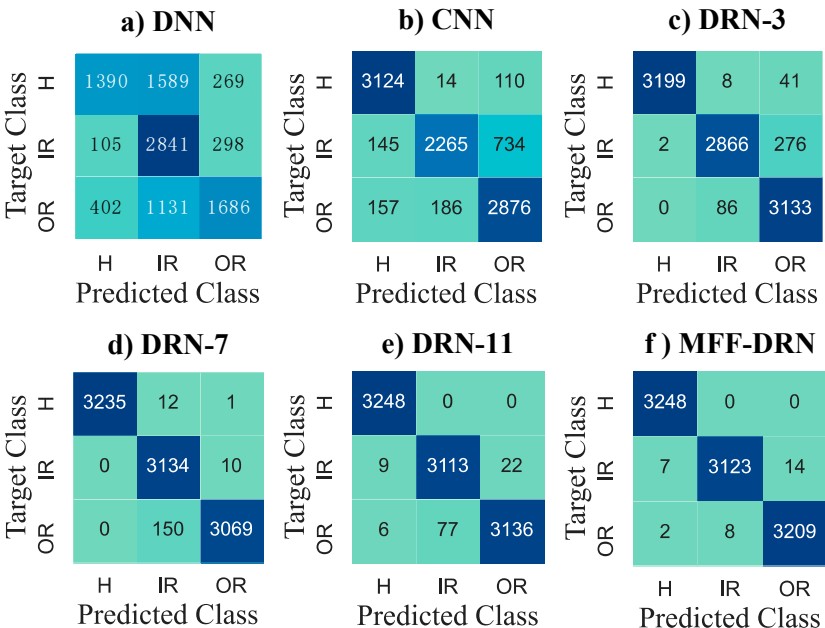

**Figure 8.** Confusion Matrix Comparison in Case 1.

The comparison of MFFDRN to some advanced methods reported in the recent literature is presented in Table 5, including transfer CNN (TCNN) [44], CNN with one-dimension convolution channels (CNN-1D) [45] and ensemble CNN (ECNN) [46]. Among these models, TCNN uses 2-D images generated by the signal-to-image conversion as inputs, and the proportion of testing test is 10%. The inputs for CNN-1 are frequency spectrum images generated by fast Fourier transform and the proportion of testing set is 1/160. ECNN takes frequency spectrum images of multiple sensor signals as input and the proportion of testing set is 20%. As opposed to these methods, the proposed MFFDRN can extract data features directly from the raw signal, avoid the design of manual features, and achieve end-to-end fault diagnosis. Compared with other comparison methods, the proposed method faces greater challenges. As shown in Table 5, the mean accuracies of TCNN, CNN-1, ECNN and MFFDRN were 98.95%, 98.58%, 98.17% and 99.73%, respectively. MFFDRN outperformed all three advanced methods, which further manifests its excellent classification ability for real faults.

**Table 5.** Comparison of MFFDRN with some advanced methods in Case 1.

| Model | Input | Mean Acc (%) |
|-------|-------|--------------|
| TCNN [44] | 2-D image | 98.95 |
| CNN-1D [45] | 2-D image | 98.58 |
| ECNN [46] | Spectrum image | 98.17 |
| MFFDRN | Raw signal segment | 99.73 |

*3.2. Case Two*

3.2.1. Data Description

The MFPT dataset was utilized to further validate the performance of MFFDRN, a dataset acquired from a test bench with NICE bearings. This dataset is composed of three conditions: healthy, inner race fault and outer race fault. All faults of rolling bearings in this dataset are caused by artificial damage. It is noted that the MFPT dataset is an unbalanced dataset, which makes this task more challenging than Case 1. More information about MFPT dataset can be found in [43].

The preprocessing of MFPT was similar to the procedure in Figure 6; the difference was that signals in this subsection didn't have to be cut off. In this case, the signal segment length was 1024. Finally, 5434 signal segments were obtained and 30% were randomly selected as the testing set. The signal segments with different health conditions are shown in Figure 9.

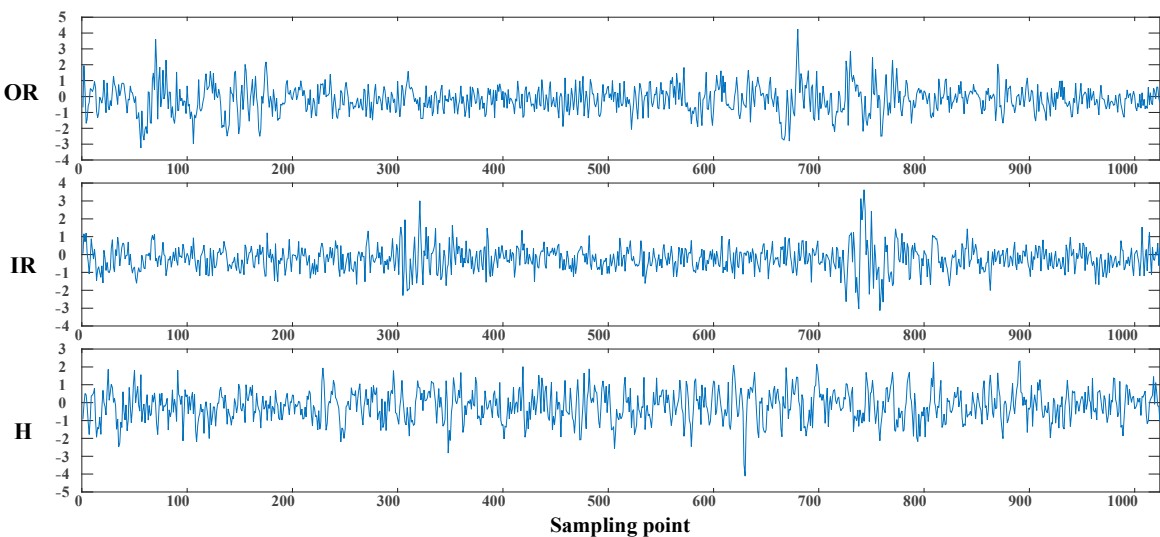

**Figure 9.** Signal segments with different conditions for Case 2.

3.2.2. Results Comparison and Analysis

DNN, CNN and DRN-3, DRN-7 and DRN-11 were also adopted as the comparison methods with MFFDRN. The model settings are the same as in Case 1 (shown in Table 3).

The experimental results are presented in Figure 10. The accuracy of MFFDRN reached 100% in the second, third, fourth and fifth repeated experiments and reached 99.94% in the first repeated experiment. The best accuracy of DRN-11 and DRN-7 were also 100% but both models were less stable than MFFDRN. The accuracy of DRN-3 was quite stable but slightly lower than MFFDRN. Table 6 records the detailed testing accuracies. Compared with DRN-7, DRN-3 and DRN-11 had a higher mean accuracy and mean *F* value. Overall, MFFDRN showed its superiority in all indicators.

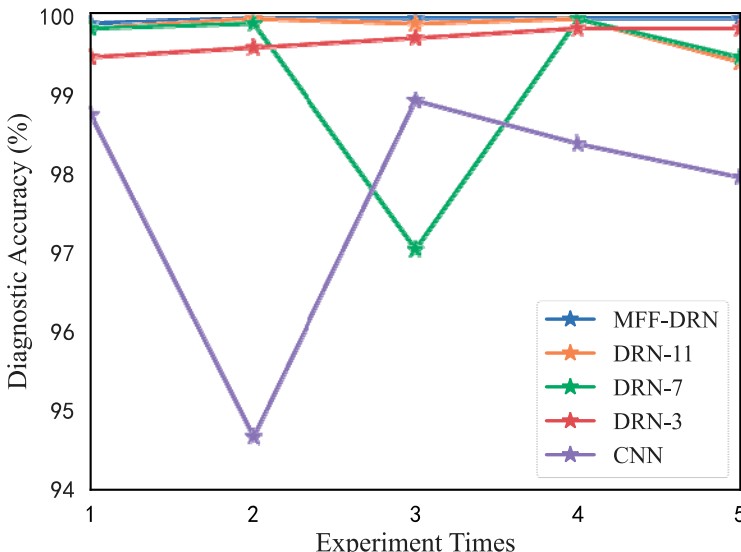

**Figure 10.** Diagnostic accuracy comparison of five methods in Case 2.

**Table 6.** Experimental results in Case 2 (%).

| Model | Max Acc | Min Acc | Mean Acc | SD | Mean *F* |
|-------|---------|---------|----------|-----|----------|
| DNN | 79.34 | 70.63 | 75.65 | 2.845 | 74.94 |
| CNN | 98.96 | 94.67 | 97.76 | 1.581 | 97.48 |
| DRN-3 | 99.88 | 99.51 | 99.73 | 0.143 | 99.75 |
| DRN-7 | 100 | 97.06 | 99.28 | 1.123 | 99.12 |
| DRN-11 | 100 | 99.45 | 99.85 | 0.207 | 99.82 |
| MFFDRN | 100 | 99.94 | 99.99 | 0.025 | 99.99 |

Figure 11 shows the classification results of the last repeated experiment. It is easy to see that the dataset is unbalanced; this is the reason why the DNN mistakenly identifies more than half of the healthy samples as an outer race fault. The CNN based models perform well on this dataset. Except for MFFDRN, all other models have some misjudgments, which reveals the superiority of the proposed model.

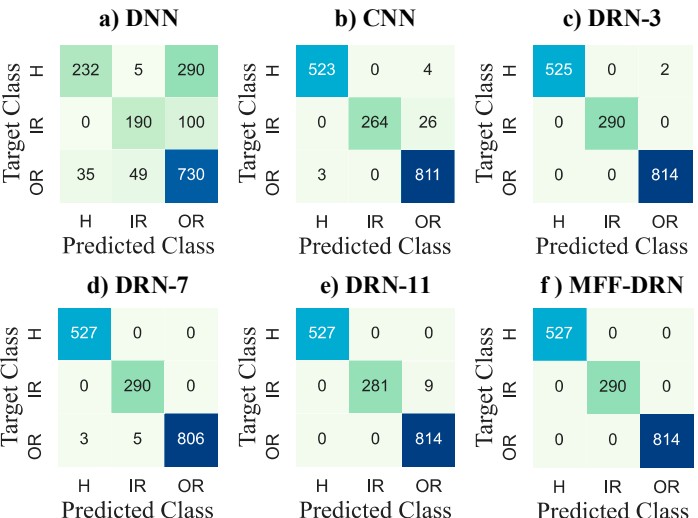

**Figure 11.** Confusion Matrix Comparison in Case 2.

In Table 7, the MFFDRN is compared with ST-CNN [47], LCNN [48], SNN [49] and local binary CNN (LBCNN) [50]. Among these models, inputs for the ST-CNN are time-frequency pictures generated by the S-transform algorithm and the proportion of testing

set is 15%. The dataset is processed to be balanced for LCNN and the proportion of testing set is 20%. The SNN uses features generated by local mean decomposition (LMD) and the proportion of testing set was 30%. The mean accuracy of 1-D CNN, LCNN, SNN and MFFDRN was 99.50%, 99.92%, 99.54% and 99.99%. The fact that MFFDRN reached the highest accuracy proves its extraordinary classification ability on the unbalanced dataset.

**Table 7.** Comparison of MFFDRN with some advanced methods in Case 2.

| Model | Input | Mean Acc (%) |
| --- | --- | --- |
| ST-CNN [47] | S-transform image | 99.50 |
| LCNN [48] | Raw signals (Balanced) | 99.92 |
| SNN [49] | Local mean decomposition feature | 99.54 |
| LBCNN [50] | Wavelet transform image | 99.56 |
| MFFDRN | Raw signal segment (Unbalanced) | 99.99 |

## 4. Conclusions

This paper proposes a novel multiscale feature fusion deep residual networks for the fault diagnosis of rolling bearings, which contains multiple multiscale feature fusion blocks and a multiscale pooling layer. The multiple multiscale feature fusion block is designed to automatically extract the multiscale features from raw signals, and further compress for higher dimensional feature mapping. The multiscale pooling layer is constructed to fuse the extracted multiscale feature mapping. Two famous rolling bearing datasets are adopted to evaluate the diagnostic performance of the proposed model. The comparison results show that the diagnostic performance of the proposed model is superior to both several popular models and to other advanced methods in the literature.

There may be strong signal interference in actual industrial applications. How to effectively remove the interference signal to achieve accurate diagnosis is a question that still needs further research. In the future, we will try to equip the proposed model with noise removal components to enable it to perform fault diagnosis tasks in complex noise environments.

**Author Contributions:** Author Conceptualization and writing—original draft, formal analysis, resources X.W.; review and editing and validation, H.S. and H.Z. All authors have read and agreed to the published version of the manuscript.

**Funding:** This research is supported by the National Natural Science Foundation of China (Grant No.52075202).

**Data Availability Statement:** Not applicable.

**Acknowledgments:** The authors thank Society for Machinery Failure Prevention Technology and Paderborn University for providing free access to the bearing vibration experimental data on their website.

**Conflicts of Interest:** The authors declare no conflict of interest.

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
