# Peer review of "Fault Diagnosis for Rolling Bearings Based on Multiscale Feature Fusion Deep Residual Networks"

_electronics, doi:10.3390/electronics12030768_

Round 1

Reviewer 1 Report

1) The paper calls intelligent fault diagnosis, however, in 2005 (IEEE Transactions on Industrial Electronics 62 (6), 3757 - 3767,  IEEE Transactions on Industrial Electronics 62 (6), 3768 - 3774), the seminal papers categorize the fault diagnosis approaches as model-based fault diagnosis, signal-based fault diagnosis and knowledge-based fault diagnosis, widely accepted by the community. Why would you like to use the term intelligent fault diagnosis? Where does this term come from?

2)  The innovation of the paper is ambiguous, which seems to be an application of the existing approaches. For the data-driven and machine learning approach, it is hard to identify the innovation as all the approaches seem to be existing. 

3) In (4), H(x)=f(x)+x. Not sure why one use this form. Actually, it can be expressed as H(x)=g(x) in a compact expression

4) Data are open sources? Did you use your own data from your experimental work?

5) You focus on bearing fault diagnosis. The journal is Electronics. How do you feel your paper suits this journal? 

Reviewer 2 Report

The author proposes a multiscale feature fusion deep residual network for intelligent fault diagnosis of rolling bearings. The authors propose a new multiscale feature fusion deep residual network (MFFDRN) for IFD of rolling bearings, which consists of multiple multiscale feature fusion blocks (MFF blocks) and multi-scale pooling layers (MPL). There are experimental results in the research results, the research results are reasonable and practical. However, the manuscript can be improved on the following points:

1.   The bibliography should be significantly expanded. The reviewer suggests adding a literature review with the latest references. The latest references are from 2020. It is recommended to add references in MDPI.

2.  In 3.1.2. Comparative analysis of the results, it is suggested that the authors add an analysis and description of the reasons for the advantages of MFFDRN.

3.  Table 3 shows the training settings for all models. However, the reason for the setting is not explained. It is suggested that the author increase the results of changing different parameters, or discuss how to optimize the parameters.

4. Table 5 shows the comparison of MFFDRN with some advanced methods in Case 1. However, the comparison standards are not the same. The input of MFFDRN is the Raw signal segment, and the input of other methods is an image, so the comparison is not fair. It is recommended that the author find other methods in Raw signal segments for comparison. Case 2 in Table 7 also has the same problem.

5. The authors show signal segments with different health statuses in Case 1 in Figure 6. OR indicates an outer-race failure, IR indicates an inner-race failure, and H indicates a healthy sample. But readers will be curious about images of the bearings, and it is recommended that the authors show images of bearing outer race failures, bearing inner race failures, and healthy bearings.

6. Figure 7 shows the diagnostic accuracy comparison of the five methods in Case 1 at the experimental times. It is recommended to add a description of the definition of the experimental times and a discussion of the results.

7. Figure 8 Confusion Matrix Comparison in Case 1. The authors are advised to confirm that the 6 methods compare with the same number, a) 3106 for DNN healthy samples but f) 3248 for MFF-DRN healthy samples.

8.  Figures 8 and 11 show the mixing matrices in Case 1 and Case 2, suggesting that the authors add results and discussions in cross-validation.

9. It is recommended that the authors' approach increases the limitations of the study in the conclusion.

Round 2

Reviewer 1 Report

Thanks for the responses from the authors. However, I cannot see any changes and update according to my comments. There are so many relevant papers on fault diagnosis for rolling bearing by using data driven approaches and machine learning techniques, so the research motivation and innovation of the paper is ambiguous. Some key references on fault diagnosis are missing. I am not convinced by the quality of the paper.. 

Reviewer 2 Report

The authors' effort in manuscript revision is greatly appreciated. The manuscript is reasonable and well-presented. The manuscript has been sufficiently improved for publication in Electronics.

Round 3

Reviewer 1 Report

1) Thanks for the quick response and revision. Although some papers used ``intelligent fault diagnosis'', but they are generally in low ranking journals. This term ``intelligent diagnosis'' is not widely accepted in fault diagnosis community. As a result, I suggest the authors avoid using the term ``intelligent fault diagnosis''. Just simply delete ``intelligent'' in the title. 

2) Avoid using the short words in the abstract. 

3) In table 1, you use * and x. You should clearly indicate their differences. 

4)  Federated learning approach has been a popular deep-learning based diagnosis approach, such as  ``Class-Imbalance Privacy-Preserving Federated Learning for Decentralized Fault Diagnosis With Biometric Authentication'', Lu et al (2022). Some discussions are encouraged. 

5) In conclusions, try to minimize using short words to enhance the interests from the readers. 

Round 4

Reviewer 1 Report

The paper is ready to be accepted.